# Endometrial Carcinoma: Immune Microenvironment and Emerging Treatments in Immuno-Oncology

**DOI:** 10.3390/biomedicines9060632

**Published:** 2021-06-02

**Authors:** Sandrine Rousset-Rouviere, Philippe Rochigneux, Anne-Sophie Chrétien, Stéphane Fattori, Laurent Gorvel, Magali Provansal, Eric Lambaudie, Daniel Olive, Renaud Sabatier

**Affiliations:** 1Immunomonitoring Department, Institut Paoli-Calmettes, 13009 Marseille, France; roussetrouvieres@ipc.unicancer.fr (S.R.-R.); rochigneuxp@ipc.unicancer.fr (P.R.); anne-sophie.chretien@inserm.fr (A.-S.C.); stephane.fattori@inserm.fr (S.F.); laurent.gorvel@inserm.fr (L.G.); lambaudiee@ipc.unicancer.fr (E.L.); daniel.olive@inserm.fr (D.O.); 2Department of Surgical Oncology, Institut Paoli-Calmettes, 13009 Marseille, France; 3Predictive Oncology Laboratory, CRCM, Inserm U1068, CNRS U7258, Institut Paoli-Calmettes, Aix Marseille University, 13009 Marseille, France; 4Department of Medical Oncology, Institut Paoli-Calmettes, 13009 Marseille, France; provansalm@ipc.unicancer.fr; 5Team Immunity and Cancer, CRCM, Inserm U1068, CNRS U7258, Institut Paoli-Calmettes, Aix Marseille University, 13009 Marseille, France

**Keywords:** endometrial carcinoma, immune micro-environment, immune checkpoints inhibitors, microsatellite instability, mismatch repair deficiency

## Abstract

Endometrial cancer (EC) can easily be cured when diagnosed at an early stage. However, advanced and metastatic EC is a common disease, affecting more than 15,000 patients per year in the United Sates. Only limited treatment options were available until recently, with a taxane–platinum combination as the gold standard in first-line setting and no efficient second-line chemotherapy or hormone therapy. EC can be split into four molecular subtypes, including hypermutated cases with POLE mutations and 25–30% harboring a microsatellite instability (MSI) phenotype with mismatch repair deficiency (dMMR). These tumors display a high load of frameshift mutations, leading to increased expression of neoantigens that can be targeted by the immune system, including (but not limited) to T-cell response. Recent data have demonstrated this impact of programmed death 1 and programmed death ligand 1 (PD-1/PD-L1) inhibitors on chemo-resistant metastatic EC. The uncontrolled KEYNOTE-158 and GARNET trials have shown high response rates with pembrolizumab and dostarlimab in chemoresistant MSI-high tumors. Most responders experiment long responses that last more than one year. Similar, encouraging results were obtained for MMR proficient (MMRp) cases treated with a combination of pembrolizumab and the angiogenesis inhibitor lenvatinib. Approvals have, thus, been obtained or are underway for EC with immune checkpoint inhibitors (ICI) used as monotherapy, and in combination with antiangiogenic agents. Combinations with other targeted therapies are under evaluation and randomized studies are ongoing to explore the impact of ICI-chemotherapy triplets in first-line setting. We summarize in this review the current knowledge of the immune environment of EC, both for MMRd and MMRp tumors. We also detail the main clinical data regarding PD-1/PD-L1 inhibitors and discuss the next steps of development for immunotherapy, including various ICI-based combinations planned to limit resistance to immunotherapy.

## 1. Introduction

In 2020, endometrial cancer (EC) was the fourth most common cancer in women, with an incidence of 382,069 new cases and 89,929 deaths worldwide in 2018 [1,2]. EC mostly affects post-menopausal women (68 years of age, on average). In industrialized countries, most patients are diagnosed in a localized stage, with a favorable prognosis (5-year overall survival: 80%) and are treated with a hysterectomy, with or without adjuvant therapy [3]. However, for patients with advanced disease, with lymph node invasion or metastasis (peritoneal or visceral), the 5-year overall survival is only 50% and 20%, respectively [4]. In advanced endometrial cancer, therapeutic options are limited: in first-line setting, a taxane–platinum combination is the gold standard, but no standard second-line treatment (chemotherapy or endocrine therapy) is available [5]. Furthermore, systemic chemotherapies are not always feasible due to patient comorbidities and performance status after platinum failure.

Endometrial cancers are broadly classified into two groups: type I endometrioid tumors are linked to estrogen excess, obesity, hormone-receptor positivity, and favorable prognosis compared with type II, primarily serous, tumors that are more common in older, non-obese women and have a worse outcome [6,7,8]. Moreover, the FIGO stage, the histological type, the pathological grade (both gathered in the old-fashioned type 1/type 2 classification), hormone receptors expression, and the presence of vascular emboli stratify prognostic groups and guide complementary treatments (classification ESMO–ESGO–ESTRO) [3]. However, the prognostic value of these classifications remains suboptimal, in particular due to the heterogeneity of tumors grouped together within the same histological type.

Molecular classification of EC based on The Cancer Genome Atlas Project (TCGA) [9], called the Proactive Molecular Risk classification tool for endometrial cancers (ProMisE) identified four classes of EC based on genomic characterization [10]: (i) ultramutated EC (harboring somatic mutations in the proofreading exonuclease domain of the DNA replicase POLE) are tumors with the highest rate of mutations and neo-antigens and the best prognosis; (ii) microsatellite instability-high (MSI-H) genotype (hypermutated) present a defect in the mismatch repair (MMR) pathway: the insertion or deletion of repeated units during DNA replication are no longer corrected by the proteins MLH1, MSH2, MSH6, and PMS2; (iii) copy number low tumors (most low grade endometrioid) have a moderate rate of mutations and exhibit both a low somatic copy number variation number (SCNAs) and a wild typeTP53 gene; (iv) copy number high tumors (serous-like) have TP53 mutations and present the lowest rate of mutations and a very large number of SCNAs.

This molecular classification was very recently incorporated into the ESGO recommendations [11]. However, this molecular classification alone does not explain the different responses to systemic therapies. A better description of the tumor immune micro-environment could refine the prognosis and help define new targets for immuno-oncological therapies [11].

## 2. Rationale for Targeting the Immune Microenvironment in Endometrial Cancer

### 2.1. Tumor-Infiltrating Lymphocytes According to Molecular Subtypes

Similarly to many tumor types (melanoma, lung, and colorectal carcinoma), the frequency of lymphocytes infiltrating tumor and peritumoral areas is correlated with the risk of recurrence in endometrial cancer. Kondratiev et al. demonstrated by immunohistochemistry (IHC) that a number of CD8 + LTs > 10 per field (at ×40 magnification) found in the peritumoral zone is an independent prognostic factor associated with improved survival [12]. However, the immune infiltrate differs among molecular subtypes of EC.

Interestingly, tumor-infiltrating CD8+ T lymphocytes within cancer cell nests are particularly abundant in MSI tumors (30% of 123 EC samples analyzed by IHC) [13]. These findings were confirmed by another study by Pakish et al. in MSI-high tumors (*n* = 60) demonstrating that immune cells were more present in stroma of MSI-H EC compared with microsatellite stable (MSS) cases, including granzyme B+ cells, activated T-cells (CD8+ granzyme B+), and PD-L1+ cells [14]. Specifically, inherited Lynch syndrome MSI-H EC had increased CD8+ cells and activated T-cells in stroma, with reduced macrophages in stroma and tumor compared with sporadic MSI-H EC [14].

Within the MSI-H subtype, immunotherapy response is associated with higher rates of tumor infiltrating immune cells, both in EC and other MSI-H tumors types [15]. MSI-H and non-MSI-H colorectal tumors exhibited distinct levels of infiltration and immune phenotypes. Surprisingly, there was no significant difference between MSI-H and non-MSI-H endometrial tumors. Regardless of cancer type, the abundance of tumor infiltrating immune cells was an independent prognostic factor, with better accuracy than MSI-H status. The authors conclude that the study of immune infiltrate is a fundamental biomarker for predicting response to immunotherapy treatments.

POLE hypermutated tumors (7–12% of EC) also had an important TILs infiltration, with an overexpression of genes involved in the cytotoxic functions of TILs, in particular T-bet, Eomes, interferon γ (IFNγ), perforin, and granzyme B, and markers of exhaustion markers on TILS, consistent with chronic exposure to neo-antigens [16]. In silico analysis confirmed that POLE-mutant cancers are predicted to display more antigenic neoepitopes than other EC, providing a potential rational for POLE immunogenicity [16].

### 2.2. The PD1/PD-L1 Axis in Endometrial Cancer

Endometrial cancer cells and tumor microenvironment are able to modulate the immune response. Among gynecological cancers, EC displays the highest overexpression of programmed cell death 1 (PD-1, CD279) and programmed cell death ligand-1 (PD-L1, CD274): 40–80% for endometrioid cancers, 10–68% for serous tumors, 23–69% for clear cells tumors, respectively [17,18]. PD-1 is a cell surface protein encoded by the *PDCD1* gene and expressed in particular on the surface of activated B and T lymphocytes [19]. The PD-1 pathway is a negative feedback system that controls the cytotoxic activity of lymphocytes in order to prevent autoimmune reactions (Figure 1).

Its major ligand PD-L1 is constitutively expressed at a low level on antigen-presenting cells (dendritic cells, macrophages and B cells) and is upregulated in these cells after their activation as well as in activated T cells and in various cancer cells [20,21,22]. PD-L1 is regulated by many inflammatory cytokines, including IFNγ, GM-CSF, LPS, IL-4, and IL-10 [23,24]. In tumors, PD-L1 expression has been abundantly detected and is often associated with a poor prognosis [25,26]. The upregulation of PD-L1 is modulated by CD8 + T cells and IFNγ. Therefore, PD-L1 expression could be viewed as a negative feedback loop dependent on an infiltrating immune response [27]. PD-L1 is expressed in 92% of endometrial cancers. High PD-L1 expression is associated with advanced tumor stage and poor tumor differentiation; however, unlike what is usually observed in other solid tumors, PD-L1 does not appear as a prognostic factor in endometrial cancers [28].

On the other hand, the PD-L2 expression is much more restricted. It is mainly expressed on antigen presenting cells but expression can be induced on several other immune and non-immune cells depending on environmental stimuli [26]. PD-L2 has moderate-to-high expression in triple negative breast cancer and gastric cancer and low expression in renal carcinoma [29]. PD-L2 is expressed at low levels within endometrial tumors, but at higher rates in serous tumors. More data are needed to better understand its role in immune response before confirming it can be considered a good candidate to target in this tumor type [30,31].

Modulation of the immune response thus appears to be different within molecular subtypes. Willvonseder et al. demonstrated greater infiltration by TILs in high-grade tumors compared to low-grade tumors, as well as in the POLE and MSI-H subgroups [32]. The greater infiltration of ultramutated POLE and MSI-H tumors is accompanied by overexpression of PD-1 and PD-L1 [33]. Likewise, the immune microenvironment of MSI-H endometrial tumors harbors more activated CD8 + T-cells and PD-L1 + cells in MSI-H vs. MSS [14]. In a large cohort of 183 EC, Kim et al. showed that a high level of PD-L1 + T-cells was significantly associated with a shorter PFS predominantly in MSS tumors [34].

### 2.3. Other Immune-Response Related Features in Endometrial Cancer

Cytotoxic T lymphocyte antigen-4 (CTLA-4), lymphocyte activation gene-3 (LAG-3), and IDO may also be upregulated in POLE tumors. In mutated TP53 tumors, infiltration by regulatory T lymphocytes is an independent prognostic factor. Analysis of immune populations by multiplexed IHC in 460 endometrial cancers stratified according to the four molecular subtypes showed a profound variation in the immune response between the molecular subtypes of endometrial cancer, but also within them [35]. Even though POLE and MSI-H tumors are the most immune-responsive, some serous-like and copy-number low tumors (defined according to IHC) also harbored strong immune responses. Besides regulatory T-cells, myeloid-derived suppressor cells (MDSCs) are also differentially expressed in EC. MDSCs are immature myeloid cells with immune suppressive action based on L-arginine depletion in the tumor microenvironment, causing T-cell receptor down-regulation [36]. MDSC levels are higher in EC than in normal endometrium [30], increase with advanced stage disease, and are associated with poor prognosis and poor response to cytotoxic treatments [37]. Tumor-associated macrophages (TAMs) are also involved in EC immune escape. EC cells expressing CD47 interact with the SIRPα macrophages-inhibiting signal. TAMs identified in EC are predominantly M2 macrophages with reduced phagocytosis properties contributing to EC progression and immune-suppression [38,39].

These points suggest that patient selection for immunomodulatory treatments may not be limited to MMRd to achieve the highest sensitivity and specificity. However, MMRd analysis by IHC remains easier to achieve and a cheaper way to predict PD1/PD-L1-inhibitors efficacy in EC, and has been widely used in recent clinical trials evaluating these drugs.

## 3. Immune Checkpoint Inhibitors in EC. PD-(L)1 Inhibitors as Backbone of all Strategies under Investigation

Until recently, cytotoxic chemotherapies were the gold standard for metastatic endometrial cancer treatment, whatever the line of treatment. The first-line regimen for advanced EC is still the combination of carboplatin and paclitaxel with overall response rates (ORR) of 50 to 60%, median progression-free survival of one year, and a median overall survival slightly of above three years [40,41]. After platinum failure, mono-chemotherapies are routinely used with poor results. For instance, doxorubicin and paclitaxel, the most prescribed second-line regimen, only offers a 4-month median PFS and a 12-month median OS [42]. Therefore, recurrent EC is a relevant clinical need and new therapies with innovative mechanisms of action have been explored to improve the outcome of patients with advanced EC. Among them, immunotherapy seems to be the most promising.

### 3.1. Clinical Trials Exploring PD-(L)1 Inhibitors as Monotherapy for Recurrent EC

As shown above, an MMRd phenotype/MSI-H genotype or POLE mutations can be predictive factors of ICI efficacy. Several research programs have thus explored the impact of ICI in these subsets. We can split trials in two categories. Studies that included less than 100 patients with metastatic or advanced endometrial cancer, mainly assessing PD-L1 inhibitors impact; and those with more than 100 patients, focused on PD-1 inhibitor large evaluations (Table 1). PD-1 inhibiting monoclonal antibodies limit the interaction of PD1 expressed by T-cells with its ligands (PD-L1 and PD-L2) upregulated in cancer cells. This inhibits the negative feedback loop resulting in the activation of anti-tumor immune response. PD-L1 specific antibodies only avoid PD-1/PD-L1 combination, resulting in a similar immune effect, but might induce less immune toxicity, notably pneumonitis.

Atezolizumab was administered, in a multi-cohort phase 1 study, to 15 patients with advanced/recurrent uterine cancer naive of anti-PD-(L)1 therapies [43]. After a dose escalation part, patients received atezolizumab at the dose of 15 mg/kg or the fixed dose of 1200 mg IV every three weeks for 16 cycles (one year) or until progression, unacceptable toxicity, or withdrawal. Treatment efficacy was analyzed according to PD-L1 expression on immune cells as defined by the Ventana PD-L1 SP142 IHC assay. Patients with less than 5% of positive IC (IC0 or IC1) and patients with 5% or more positive IC (IC2 and IC3) were separated. Overall response rate (ORR) was 13% (two responders out of 15 patients). These two responders were IC2/3 (among five cases). None of the ten IC0/1 patients had objective responses. It is worth noting that one responder had a MHI-high/TMB-high tumor (the other was not evaluable) whereas none of the non-responders had microsatellite instability or high tumor mutational burden. Median PFS was 4.2 months in the IC2/3 group vs. 1.4 months in the IC0/1 subset. Statistical evaluation of the significance of these results could not be performed due to the very small sample size.

Nivolumab (240 mg IV every 2 weeks until progression or unacceptable toxicity) was explored in a Japanese phase 2 multicohort open-label trial, including 23 unselected EC [44]. Results have been presented at the ASCO meeting in 2020. ORR was 23%, with a median PFS of 3.4 months and a 12-month OS of 42%. PD-L1 expression was not predictive of response and both patients with MSI-H tumors displayed partial response.

A non-randomized phase 2 trial enrolled previously treated recurrent EC in two cohorts exploring avelumab efficacy [45]. A dMMR/POLEmut cohort of 15 patients and a pMMR/non-POLEmut cohort of 16 patients. They received avelumab at the dose of 10 mg/kg IV every two weeks for a maximum of 24 months. Most of objective responses were observed in the dMMR cohort (26.7%) with one complete response (CR) and three partial responses (PR). The median PFS was 4.4 months in this subset with a 6-month PFS rate of 40%. Only one patient responded in the pMMR cohort, for a 6-month PFS rate of 6.3%. Exploratory analyses did not identify a predictive biomarker. Among the 25 tumors with PD-L1 expression evaluated using the tumor proportion score (considered positive if TPS ≥ 1%), seven (28%) were PD-L1 positive. However, all patients with objective responses in both dMMR and pMMR cohorts had PD-L1–negative tumors. Tumor mutational burden and tumor infiltrating lymphocytes were also not correlated to response.

The phase 2 non-randomized PHAEDRA study enrolled 71 patients (36 dMMR and 35 pMMR) with advanced EC [46]. They received durvalumab 1500 mg IV every 4 weeks. Patients with dMMR tumors should have received less than four prior lines of treatment but could have been treatment naïve in the metastatic setting, whereas patients with pMMR disease had to be pre-treated. ORR was 47% in the dMMR population (6 CR plus 11 PR) vs. 3% in the pMMR subgroup. Median PFS was 5.5 months for dMMR cases with a 1-year overall survival rate of 71% vs. 1.8 months and a 51% 1-year survival rate for pMMR cases. Sensitivity to durvalumab seemed to be higher for treatment-naïve patients with a 57% ORR when durvalumab was given as first-line treatment vs. 38% when the PD-L1 inhibitor was administered in the second-line setting. The PD-L1 combined positive score (22C3 antibody) was negative in most responders.

Pembrolizumab (anti-PD-1) impact was first assessed in a cohort of the multitumor phase I KEYNOTE-028 trial [47]. Twenty-four patients with PD-L1 positive EC received pembrolizumab alone. Three of them (13%) had objective responses, including one case with POLE mutation. The Keynote-158 trial was a multi-cohort phase 2 study that enrolled patients with previously treated metastatic solid tumors. Patients with EC could be included in two cohorts: cohort D (N = 107) with all EC and cohort K with MSI-H solid tumors [48,49]. Forty-nine MSI-H cases (21% of the whole study population) were enrolled in both cohorts. Patients received pembrolizumab (PD-1 inhibitor) every three weeks for a maximum of 35 cycles. ORR was 11.2% in cohort D and 57.1% for MSI-H tumors (45.5% of 11 patients in cohort D and 60.5% of 38 patients in cohort K). Among MSI-H cases, less than 25% (11 cases) displayed progression as best responses. Median duration of response was not reached with 89% of responders still in response at one year. One-year PFS and OS rates were 58.4% and 73.5%, respectively. According to these results, and that of other cohorts of this study, pembrolizumab obtained FDA approval for all unresectable or metastatic dMMR/MSI-H solid tumors, irrespective of localization, with progression following treatment, and for which there were no satisfactory alternative treatment options [55]. Response according to PD-L1 expression was not described.

The GARNET trial is an open-label uncontrolled multi-cohorts phase 1 trial exploring the impact of dostarlimab (PD-1 inhibitor) in various tumor types. The A1 (dMMR) and A2 (pMMR) cohorts of the expansion phase enrolled patients with previously treated EC [50,56]. They receive dostarlimab at the dose of 500 mg IV every 3 weeks for four doses, then 1000 mg IV every 6 weeks until disease progression. Efficacy data have been presented for 103 dMMR and 142 pMMR EC. ORR were 44.7% and 13.4% in dMMR and pMMR cohorts, respectively. Disease control rate was 57.3% and 35.2% in each cohort. Median duration of response has not been reached in both cohort, with 89% of responders still in response in the dMMR subgroup. PFS and OS data are not yet available for this trial, as well as efficacy according to PD-L1 expression. FDA and EMA approvals for dMMR recurrent EC were obtained in April 2021.

No new safety signal was identified in these trials, without any treatment-related death. Eleven to nineteen grade 3 treatment-related adverse events were observed. Immune related adverse events (AEs) were mainly dysthyroidism and digestive disorders.

### 3.2. PD-(L)1 Inhibitors-Based Combinations

Due to the complexity of immune response activation and the various mechanisms leading to resistance to PD-(L)1 inhibitors, combination strategies were developed to obtain synergistic benefits or to reduce primary or secondary resistance. In order to inhibit other immune checkpoints, combos with CTLA-4, TIGIT, IDO, and PVRIG are under early clinical evaluations (NCT03015129, NCT04570839, NCT04106414, NCT03667716) [57], and future data will provide insights about their clinical utilities in this setting.

The most advanced combinations explore angiogenesis and PARP inhibitors in the recurrent setting, and chemotherapy in the first-line setting.

#### 3.2.1. Angiogenesis Inhibitors

Angiogenesis-inhibiting drugs are described as having a synergistic effect with ICI, by decreasing hypoxia, which is correlated with myeloid cell activation; enhancing T cells spreading to the tumor microenvironment; and favoring lymphocyte activation [58,59]. Combinations with ICI have, thus, been developed to limit primary resistance to immunotherapy.

Lenvatinib is an oral multityrosine kinase inhibitor targeting VEGFR among other proteins. Firstly developed as monotherapy in thyroid cancer and hepatocellular carcinoma, it is currently widely explored in other solid tumors in combinations with ICI. Concerning EC, the first evidence of activity has been observed in association with pembrolizumab in the KEYNOTE-146 study, a phase Ib/II multicohort trial [51]. One hundred and eight previously treated patients with EC were enrolled, including 11 with dMMR tumor and 94 MMRp cases. They received pembrolizumab 200 mg every 3 weeks and lenvatinib 200 mg once daily. ORR at week 24 (primary endpoint) was 38% in the whole population, 36.2% for MSS tumors, and 63.6% in the MSI-H subset. Seven (7.3%) patients experienced complete responses and 28 (29.8%) partial responses in the MSS cohort. Responses were observed in all histologic subtypes. Median duration of response was not reached in this cohort. Outcome seemed better than in historical cohorts with 7.4 months and 16.4 months of median PFS and OS, respectively. PD-L1 status was determined by the 22C3 assay, with 49% of PD-L1 positive tumors. Treatment efficacy did not depend on PD-L1 expression, as 35.8% of patients with PD-L1-positive tumors and 39.5% of patients with PD-L1-negative tumors had objective responses, respectively. Concerning safety, the combination led to more adverse events than pembrolizumab alone with grade 3–4 AEs for 66.9% of patients. Immune-related AEs were observed for more than half (57.3%) of the patients, but mostly concerned dysthyroidism (47.6%). Angiogenesis inhibition-related toxicities were frequent with 32.4% grade 3–4 hypertension. Twenty-two patients (17.7%) discontinued at least one of the drugs because of AEs, and dose interruptions were necessary for 70.2%. These results led the FDA to grant accelerated approval for this combination for the treatment of patients with MSS/MMRp advanced endometrial carcinoma, who have disease progression following prior systemic therapy, but are not candidates for curative surgery or radiation. A phase trial assessing this combination to physician’s choice chemotherapy (paclitaxel or doxorubicin) in the same setting is ongoing ([52], NCT03517449). Patients are stratified according to their MMR status; and MMRp cases are stratified according to ECOG performance status, geographic region, and priori history of pelvic radiation. Interim results of co-primary endpoints (PFS and OS) have been presented very recently [Makker V, et al. Abstract 11512. Society of Gynecologic Oncology Annual Meeting on Women’s Cancer; 2021]. Median PFS was 7.2 months for the combination versus 3.8 months in the chemotherapy arm in the whole population (HR = 0.56 (95CI 0.47–0.66), *p* < 0.0001) and 6.6 versus 3.8 in the MMRp subset (HR = 0.50 (95CI 0.40–0.72), *p* < 0.0001). Results were similar concerning OS with more than 5 months of median OS improvement in the experimental arm for MMRp tumors (17.4 vs. 12.0 months; HR = 0.68 (95CI 0.56–0.84), *p* = 0.0001). Moreover, the pembrolizumab–lenvatinib combination is also explored in the first-line setting in the ongoing ENGOT-en9 phase III study [60]. Newly diagnosed stage III–IV EC are randomized (1:1 ratio) between the combo and the carboplatin-paclitaxel regimen. Co-primary endpoints are PFS and OS (Table 2).

In a 2/1 ratio randomized phase II study for pretreated advanced EC, 76 patients (with only two MSI-H cases) received nivolumab alone or in combination with cabozantinib [53]. Median PFS was statistically higher in the combination arm: 5.3 months versus 1.9 months with nivolumab alone. An exploratory cohort with 9 carcinosarcoma and 20 patients previously treated with ICI was also presented. Only one patient with carcinosarcoma responded to treatment and six in the prior ICI subgroup. Digestive disorders (47.2%), transaminases increase (44.4%, and fatigue (8.9%) were the most frequent adverse events in the cabozantinib arm.

Others multikinase angiogenesis inhibitors (lucitanib, anlotinib) are under investigation in association with anti-PD-1 agents in a single-arm phase II trials for pretreated EC (NCT04042116, NCT04157491).

Efficacy of bevacizumab, a well described VEGFR inhibiting monoclonal antibody, has been widely assessed for EC treatment in the last 10 years. Used as monotherapy after platinum-failure, only few (13%) objective responses were observed, with a 6-month PFS of 40% [64]. Similar results were obtained in combination with the mTOR-inhibitor temsirolimus [65]. In the first-line setting, addition of bevacizumab to the standard carboplatin-paclitaxel regimen did not bring any survival benefits compared to chemotherapy alone [66]. Nevertheless, due to the potential synergistic effect of this drug in combination with ICI, bevacizumab is currently explored in combination with atezolizumab in a phase II single-arm trial planned to enroll 55 patients with previously treated advanced EC (NCT03526432).

#### 3.2.2. PARP Inhibitors

PARP inhibitors were developed in the last years, mainly in several tumor types with homologous recombination deficiency, notably with BRCA1/2 mutations [67,68,69,70,71,72,73,74]. These drugs are thought to be able to enhance ICI activity via various pathways [75]. By altering DNA repair mechanisms in cancer cells, they probably enhance the number of genomic alterations (also known as tumor mutation burden) that is a surrogate marker of ICI efficacy [76]. Moreover, double-strand breaks repair decrease also leads to the ATM–ATR–Chk1 pathway, resulting in PD-L1 upregulation, which may make higher the impact of PD-L1 blockades [77]. Cancer development is frequently associated with chronic inflammation driven by interferon (IFN) production. However, PARP inhibitors also enhance IFN production via the cGAS–STING pathway [78]. This pathway is activated by the accumulation of cytoplasmic double-strand DNA. STING activation induces type I IFN synthesis TBK1 and IRF3. As type I IFN is involved in regulation of multiple immune cell types, including DCs, NK cells, and T cells, this could enhance ICI impact. Some early phase studies were launched during the last years to explore these hypotheses.

The Dana–Farber Institute is currently sponsoring a phase II multicohort trial, including 35 MSS cases receiving avelumab (PD-L1 inhibitor) in association with the PARP inhibitor talazoparib (NCT02912572). Preliminary results of this cohort were presented at the 2020 ESMO meeting [54]. Thirty-five patients with pretreated advanced EC have been included. Only three of them had partial response (ORR = 8.6%) including one of 12 serous tumors. Six-month PFS was 25.8%. Most common grade 3–4 toxicities were hematological disorders with anemia (45.7%), thrombocytopenia (28.6%), and neutropenia (11.4%). Following the same biological basis, another PARP inhibitor (niraparib) is explored in combination with dostarlimab in a Canadian cohort of 44 pretreated advanced EC (NCT03016338).

In the non-randomized phase II ENDOBAAR trial, 30 patients with previously treated advanced EC are receiving a triplet with bevacizumab, atezolizumab, and the PARP inhibitor rucaparib (NCT03694262). Efficacy of the combination will be assessed by estimating the ORR.

Patients with advanced EC harboring homologous recombination genes mutations are currently enrolled in the multicohort GUIDE2REPAIR trial [79]. This non-randomized phase II study will explore the combination of dual blockade with durvalumab and the CTLA-4 inhibitor tremelimumab with olaparib (PARP inhibitor).

#### 3.2.3. Chemotherapy

Cytotoxic agents are the corner stone of cancer treatment since the 1950s with huge successes for chemosensitive diseases, such as testicular cancer. However, despite administration of multidrug regimen or use of high-dose treatments, chemotherapy only brings a short survival improvement in most metastatic solid malignancies. Chemotherapy action is based on cell-cycle arrest induced by DNA alterations by replication or nucleotides synthesis inhibition or by mitosis inhibition. However, primary and secondary resistances nearly always occur because of emergence of new genetic and epigenetic alterations or upregulation of multidrug transporters. New paradigms appear to combine cytotoxic chemotherapies to other anticancer treatments. Concerning immunotherapy, several observations have been made suggesting that chemotherapy may enhance ICI efficacy. First, some cytotoxic drugs can induce immunogenic cell death. For instance, apoptosis resulting from the action of platinum and alkylating agents seems to be the best candidate for CD8 T lymphocytes activation via various mechanisms that are detailed elsewhere [80]. Secondly, chemotherapy can help to deplete immune response inhibiting cells, such as Tregs, MDCSs, and protumoral macrophages [61,62,63]. Third, chemotherapy may induce a homeostatic proliferation of T cells by inducing lymphopenia [81]. Lymphotoxic chemotherapy may help reshape the T-cells repertoire by favoring differentiation to tumor-killing T cells. Finally, chemotherapy can help reduce the tumor burden. Tumor volume is indeed correlated with immune response efficacy. Large tumor masses are more immunosuppressive than small cancers, and antitumor immune response is likely to be more effective on small volume tumors [82].

No relevant preliminary data have been published so far concerning the associations of immunotherapy and chemotherapy for advanced EC. However, several phase III trials are ongoing for patients treated in the first-line setting (Table 2). The NRG-GY018 study is comparing carboplatin (AUC5-6 every 3 weeks), plus paclitaxel (175 mg/m^2^ every 3 weeks), plus placebo to the same chemotherapy regimen with pembrolizumab (NCT03914612). It is of note that pembrolizumab may be continued until 5 years after inclusion. The control arm of the AtTEnd/ENGOT-en7 study is the same as above, and is compared to the combination of chemotherapy and atezolizumab (NCT03603184) [83]. Patients included in the RUBY study are treated with the same chemotherapy regimen plus dostarlimab (NCT03981796) [84]. We can also cite the Italian MITO END-3 phase 2 randomized study evaluating the same chemotherapy regimen associated with avelumab or placebo (NCT03503786). Finally, DUO-E is a phase 3 randomized trial assessing the combination of carboplatin–paclitaxel chemotherapy plus durvalumab or placebo and olaparib or placebo (NCT04269200) [85].

#### 3.2.4. Other Associations

Immune-checkpoint inhibitors are also under investigation with other anticancer therapeutics used in EC, such as radiation therapy and innovative targeted therapies. Activation of antitumor immune response by radiotherapy is an old myth known for decades as the abscopal effect [86]. There is increasing evidence that the association of radiotherapy with immunotherapy may help to boost immune response, at the irradiated site, but also in a distant manner [87,88]. Several preclinical and clinical studies are ongoing to explore efficacy and safety issues related to this combination. Concerning EC, some dedicated clinical trials have been initiated in both early stage and advanced disease. Pembrolizumab is assessed in combination with radiotherapy in dMMR high intermediate risk early stage EC in a multicenter phase III randomized study (NCT04214067). The PRIMMO study is an ongoing randomized phase II trial evaluating pembrolizumab plus hypo-fractioned radiotherapy, plus an immunomodulatory cocktail (vitamin D, curcumin, lansoprazole, aspirin, and low-dose cyclophosphamide) in patients with pretreated advanced uterine tumors (cervix or endometrial carcinoma and uterine sarcoma) [89]. Primary endpoint is ORR at week 26.

Netrin-1 is a protein overexpressed in over 80% of uterine tumors. Netrin-1 up-regulation is a mechanism to allow escape from apoptosis [90]. In early clinical studies, NP137, a monoclonal antibody that targets this protein, showed promising response rates in a not yet published first in a human study (NCT02977195). Moreover, some preclinical data suggest that it may decrease resistance to chemotherapy and ICI [91]. A phase Ib/II study was recently initiated to assess the combination of NP137 with pembrolizumab and/or chemotherapy in pretreated patients with locally advanced/metastatic endometrial carcinoma or cervix carcinoma (NCT04652076). Other combinations with investigational treatments (ataluren, mirvetuximab soravtansine) are currently evaluated in early phase trials (NCT04014530, NCT03835819).

## 4. Conclusions

There is now strong evidence that immune microenvironment modifications and immune response activation are of high importance for EC. Mismatch repair status has to be determined for all our patients. Despite its limits to perfectly identify responders to immune checkpoints inhibitors, MMR deficiency is a recognized theranostic marker for clinical management of advanced EC. Pembrolizumab and dostarlimab have shown impressive results in MMR deficient cases, and the association of pembrolizumab and lenvatinib is becoming a standard of care for pretreated recurrent MMR proficient EC. However, further advances are needed to understand primary and secondary mechanisms of resistance to immunotherapy and to implement ICI in the first-line metastatic setting and in early stage tumors.

## Figures and Tables

**Figure 1 biomedicines-09-00632-f001:**
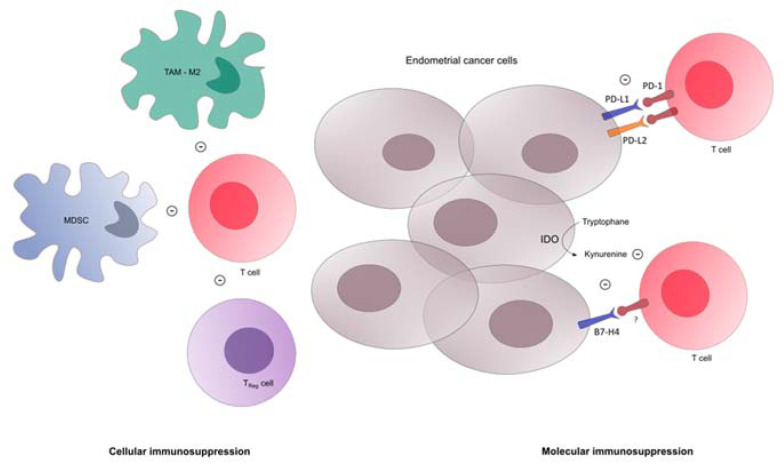
Immunosuppression in tumor microenvironment of endometrial cancer. MDSC: myeloid derived suppressor cell, Treg: T regulator lymphocyte, IDO: indoleamine-2,3-dioxygenase, TAM: tumor associated macrophage.

**Table 1 biomedicines-09-00632-t001:** Prospective clinical studies with results available.

Drugs	ICI Target	Schemes	N	ORR (%)	Median PFS (Months)	Median OS (Months)	Reference
**Monotherapy (all for pretreated advanced EC)**							
Atezolizumab	PD-L1	15 mg/kg or 1200 mg IV Q3W	15	13.3	1.4 (4.2 for IC2/3 cases)	9.6	[43]
Nivolumab	PD-1	240 mg IV Q2W	23	23	3.4	NA	[44]
Avelumab	PD-L1	10 mg/kg IV Q2W	15 dMMR 16 pMMR	26.7 6.25	4.4 6.25	Not reached 6.6	[45]
Durvalumab	PD-L1	1500 mg IV Q4W	36 dMMR 35 pMMR	47 3	5.5 1.8	Not reached 11.5	[46]
Pembrolizumab	PD-1	10 mg/kg IV Q2W	24 PDL1+	13	1.8	Not reached	[47]
		200 mg IV Q3W	107 unselected 49 MSI-H	11.2 57.1	NA 26.0	NA Not reached	[48,49]
Dostarlimab	PD-1	500 mg IV Q3W for 4 doses then 1000 mg IV Q6W	103 dMMR 142 pMMR	44.7 13.4	NA NA	NA NA	[50]
**Combinations**							
Pembrolizumab + lenvatinib	PD-1	200 mg IV Q3W + 20 mg orally once per day Single-arm ph2 in pretreated EC	94 pMMR 11 dMMR	24 W-ORR 36.2% 63.6%	7.4 for the whole set	16.7 for the whole set	[51]
		200 mg IV Q3W + 20 mg orally once per day Randomized ph3 vs. chemo in pretreated EC	697 pMMR 130 dMMR	NA	6.6 vs. 3.8 for pMMR HR = 0.60 (95CI 0.50–0.72)	17.4 vs. 12.0 for pMMR HR = 0.68 (95CI 0.56–0.84)	[52]
Nivolumab ± cabozantinib	PD-1	240 mg IV Q2W ± 40 mg orally once per day Ph 2 randomized study	36 nivo + cabo 18 nivo	25.0 16.7	5.3 1.9 *p* = 0.07 (significant)	NA	[53]
Avelumab + talazoparib	PD-L1	1200 mg IV Q3W +1 mg orally once per day Single-arm ph2 for pretreated EC	35 pMMR	8.6	6m-PFS = 25.8%	NA	[54]

Abbreviations: ICI, immune-checkpoint inhibitor; ORR, overall response rate; PFS, progression-free survival; OS, overall survival; EC, endometrial cancer; IC, tumor-infiltrating immune cell; dMMR, MMR deficient; pMMR, MMR proficient; MSI-H, microsatellite instability-high; NA, not available.

**Table 2 biomedicines-09-00632-t002:** Ongoing prospective clinical studies with combinations including immune checkpoints inhibitors.

Drugs	Study Design	N	Primary Objectives	Reference, NCT
First-line setting				
Pembrolizumab–lenvatinib vs. carboplatin–paclitaxel	Randomized ph 3	875	PFS + OS	EnGOT-en9 [57] NCT03884101
Carboplatin–paclitaxel + pembrolizumab/placebo	Randomized ph 3	220 dMMR 590 pMMR	PFS	NRG-GY018 NCT03914612
Carboplatin–paclitaxel + atezolizumab/placebo	Randomized ph 3	550	PFS + OS	AtTEND [61] NCT03603184
Carboplatin–paclitaxel + dostarlimab/placebo	Randomized ph 3	470	PFS	RUBY [62] NCT03981796
Carboplatin–paclitaxel + avelumab/placebo	Randomized ph 2	120	PFS	MITO END-3 NCT03503786
Carboplatin–paclitaxel + durvalumab/placebo + olaparib/placebo	Randomized ph 3	699	PFS	DUO-E [63] NCT04269200
Pretreated advanced EC				
Lucitanib + nivolumab	Multicohort non-randomized ph 2	227 not limited to EC	ORR	NCT04042116
Anlotinib + anti-PD-1	Non-randomized phase 2	23	ORR	NCT04157491
Atezolizumab + bevacizumab	Non-randomized phase 2	55	ORR	NCT03526432
Atezolizumab + bevacizumab + rucaparib	Non-randomized phase 2	30	ORR	ENDOBARR NCT03694262
Dostarlimab + niraparib	Non-randomized phase 2	44	CBR	NCT03016338
Durvalumab + tremelimumab + olaparib	Multicohort non-randomized ph 2 limited to HRD solid tumors	270 not limited to EC	PFS	GUIDE2REPAIR NCT04169841
Pembrolizumab + hypo-fractioned radiotherapy + immunomodulatory cocktail	Randomized phase 2	43 uterine cancer	26W-ORR	PRIMMO NCT03192059
NP137 + pembrolizumab and/or carboplatin/paclitaxel	Non-randomized phase 1b/2	240 uterine carcinoma	ORR	GYNET NCT04652076
Ataluren + pembrolizumab	Non-randomized phase 1b/2	47 EC or CCR	ORR	NCT04014530
Mirvetuximab soravtansine + pembrolizumab	Non-randomized phase 2	35 pMMR	ORR + PFS	NCT03835819

Abbreviations: PFS, progression-free survival; OS, overall survival; ORR, overall response rate; CBR, clinical benefit rate; EC, endometrial cancer; HRD, homologous recombination deficiency.

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
