# Peer review of "Endometrial Carcinoma: Immune Microenvironment and Emerging Treatments in Immuno-Oncology"

_biomedicines, 2021, doi:10.3390/biomedicines9060632_

Round 1

Reviewer 1 Report

The authors present a nice review of ongoing clinical trials and potential of PD-1 and PD-L1 inhibitors in treatment of advanced endometrial cancer both as monotherapy or in combination with other drugs. However, there are some minor changes or additions to the manuscript that should be taken into considerations:

  1. The title of the manuscript does not reflect the content, it should be more specific and mention PD-1, PD-L1 inhibitors.
  2. I would suggest adding explanation which of the drugs is a PD-1 and which PD-L1 inhibitor to the chapter 3. and also a more detailed explanation of the actions of the individual type of inhibitor.
  3. The manuscript would also benefit from additional explanation of how PD-L1 expression or PD-1 expression was analyzed in individual clinical trial and comparison between the tests.

Author Response

Dear Editor,

Please, find enclosed a revised version of our manuscript biomedicines-1239903 by Rousset-Rouviere and colleagues entitled « Endometrial Carcinoma: Immune Microenvironment and Emerging Treatments in Immuno-Oncology » submitted for publication to Biomedicines.

We thank the reviewer their additional positive and helpful comments, which have been taken into account as follows. As you will see, we have answered the questions raised by the reviewer and modified the manuscript as suggested.

Referee’s Comments to the Author and Authors’ Responses

Reviewer 1

The authors present a nice review of ongoing clinical trials and potential of PD-1 and PD-L1 inhibitors in treatment of advanced endometrial cancer both as monotherapy or in combination with other drugs. However, there are some minor changes or additions to the manuscript that should be taken into considerations:

  1. The title of the manuscript does not reflect the content, it should be more specific and mention PD-1, PD-L1 inhibitors.

We agree that most of clinical data involved PD-L1 inhibitors, we thus have changed the title of part 3 “Immune checkpoint inhibitors in EC. PD-(L)1 inhibitors as backbone of all strategies under investigation”. However, as our review is not limited to the PD(L1) axis we would like to keep the main title in this current form.

  1. I would suggest adding explanation which of the drugs is a PD-1 and which PD-L1 inhibitor to the chapter 3. and also a more detailed explanation of the actions of the individual type of inhibitor.

Details concerning the targets have been added in a new column in table 3. Concerning mechanisms of action, PD-1/PD-L1 2 axis physiology and cancer-related alterations are detailed in section 2.2. We nevertheless have added the following sentences in the beginning of the 3.1 section : “PD-1 inhibiting monoclonal antibodies limit the interaction of PD1 expressed by T-cells with its ligands (PD-L1 and PD-L2) upregulated in cancer cells. This inhibits the negative feedback loop resulting in the activation of anti-tumor immune response. PD-L1 specific antibodies only avoid PD-1/PD-L1 combination, resulting in a similar immune effect, but might induce less immune toxicity, notably pneumonitis”.

  1. The manuscript would also benefit from additional explanation of how PD-L1 expression or PD-1 expression was analyzed in individual clinical trial and comparison between the tests.

We agree that PD-L1 data will be of interest for readers. To complete our paper, we have added, for all clinical trials for which the data are known and published, the assay used for PD-L1 assessment as well as efficacy results according to this feature.

We hope that this new version will meet with your approval for publication in Biomedicines.

Sincerely yours,

Dr Renaud SABATIER

Reviewer 2 Report

The manuscript covers a wide range of topics, from the basics of ICIs for endometrial cancer to the latest clinical trials, in an easy-to-understand format. The contents are comprehensive and acceptable.

The authors need to correct reference number 82, the DUO-E trial, in which bevacizumab is not administered.

Author Response

Dear Editor,

Please, find enclosed a revised version of our manuscript biomedicines-1239903 by Rousset-Rouviere and colleagues entitled « Endometrial Carcinoma: Immune Microenvironment and Emerging Treatments in Immuno-Oncology » submitted for publication to Biomedicines.

We thank the reviewer their additional positive and helpful comments, which have been taken into account as follows. As you will see, we have answered the questions raised by the reviewer and modified the manuscript as suggested.

Referee’s Comments to the Author and Authors’ Responses

Reviewer 1

The authors present a nice review of ongoing clinical trials and potential of PD-1 and PD-L1 inhibitors in treatment of advanced endometrial cancer both as monotherapy or in combination with other drugs. However, there are some minor changes or additions to the manuscript that should be taken into considerations:

  1. The title of the manuscript does not reflect the content, it should be more specific and mention PD-1, PD-L1 inhibitors.

We agree that most of clinical data involved PD-L1 inhibitors, we thus have changed the title of part 3 “Immune checkpoint inhibitors in EC. PD-(L)1 inhibitors as backbone of all strategies under investigation”. However, as our review is not limited to the PD(L1) axis we would like to keep the main title in this current form.

  1. I would suggest adding explanation which of the drugs is a PD-1 and which PD-L1 inhibitor to the chapter 3. and also a more detailed explanation of the actions of the individual type of inhibitor.

Details concerning the targets have been added in a new column in table 3. Concerning mechanisms of action, PD-1/PD-L1 2 axis physiology and cancer-related alterations are detailed in section 2.2. We nevertheless have added the following sentences in the beginning of the 3.1 section : “PD-1 inhibiting monoclonal antibodies limit the interaction of PD1 expressed by T-cells with its ligands (PD-L1 and PD-L2) upregulated in cancer cells. This inhibits the negative feedback loop resulting in the activation of anti-tumor immune response. PD-L1 specific antibodies only avoid PD-1/PD-L1 combination, resulting in a similar immune effect, but might induce less immune toxicity, notably pneumonitis”.

  1. The manuscript would also benefit from additional explanation of how PD-L1 expression or PD-1 expression was analyzed in individual clinical trial and comparison between the tests.

We agree that PD-L1 data will be of interest for readers. To complete our paper, we have added, for all clinical trials for which the data are known and published, the assay used for PD-L1 assessment as well as efficacy results according to this feature.

Reviewer 2

The manuscript covers a wide range of topics, from the basics of ICIs for endometrial cancer to the latest clinical trials, in an easy-to-understand format. The contents are comprehensive and acceptable.

The authors need to correct reference number 82, the DUO-E trial, in which bevacizumab is not administered.

We feel sorry for the mistake. Bevacizumab has been deleted from both the text and Table 2

We hope that this new version will meet with your approval for publication in Biomedicines.

Sincerely yours,

Dr Renaud SABATIER
